# Microbial Sharing between Pediatric Patients and Therapy Dogs during Hospital Animal-Assisted Intervention Programs

**DOI:** 10.3390/microorganisms9051054

**Published:** 2021-05-13

**Authors:** Kathryn R. Dalton, Kathy Ruble, Laurel E. Redding, Daniel O. Morris, Noel T. Mueller, Roland J. Thorpe, Jacqueline Agnew, Karen C. Carroll, Paul J. Planet, Ronald C. Rubenstein, Allen R. Chen, Elizabeth A. Grice, Meghan F. Davis

**Affiliations:** 1Department of Environmental Health and Engineering, Johns Hopkins University Bloomberg School of Public Health, Baltimore, MD 21205, USA; jagnew@jhu.edu (J.A.); mdavis65@jhu.edu (M.F.D.); 2Departments of Oncology and Pediatrics, Johns Hopkins University School of Medicine, Baltimore, MD 21205, USA; rubleka@jhmi.edu (K.R.); chenal@jhmi.edu (A.R.C.); 3Department of Clinical Studies, University of Pennsylvania School of Veterinary Medicine, Kennett Square, PA 19348, USA; lredding@upenn.edu; 4Department of Clinical Sciences & Advanced Medicine, University of Pennsylvania School of Veterinary Medicine, Philadelphia, PA 19104, USA; domorris@vet.upenn.edu; 5Department of Epidemiology, Johns Hopkins University Bloomberg School of Public Health, Baltimore, MD 21205, USA; noeltmueller@jhu.edu; 6Department of Health, Behavior and Society, Johns Hopkins University Bloomberg School of Public Health, Baltimore, MD 21205, USA; rthorpe@jhu.edu; 7Department of Pathology, Division of Medical Microbiology, Johns Hopkins University School of Medicine, Baltimore, MD 21205, USA; kcarrol7@jhmi.edu; 8Department of Pediatrics, University of Pennsylvania Perelman School of Medicine, Philadelphia, PA 19104, USA; planetp@email.chop.edu; 9Department of Pediatrics, Division of Allergy and Pulmonary Medicine, Washington University in St. Louis School of Medicine, St. Louis, MO 63110, USA; rubenstein@wustl.edu; 10Department of Dermatology, University of Pennsylvania Perelman School of Medicine, Philadelphia, PA 19104, USA; egrice@pennmedicine.upenn.edu; 11Johns Hopkins Medicine, Department of Molecular and Comparative Pathobiology, Baltimore, MD 21205, USA

**Keywords:** microbiome, animal-assisted interventions, hospital-associated pathogens, patient safety

## Abstract

Microbial sharing between humans and animals has been demonstrated in a variety of settings. However, the extent of microbial sharing that occurs within the healthcare setting during animal-assisted intervention programs is unknown. Understanding microbial transmission between patients and therapy dogs can provide important insights into potential health benefits for patients, in addition to addressing concerns regarding potential pathogen transmission that limits program utilization. This study evaluated for potential microbial sharing between pediatric patients and therapy dogs and tested whether patient–dog contact level and a dog decolonization protocol modified this sharing. Patients, therapy dogs, and the hospital environment were sampled before and after every group therapy session and samples underwent 16S rRNA sequencing to characterize microbial communities. Both patients and dogs experienced changes in the relative abundance and overall diversity of their nasal microbiome, suggesting that the exchange of microorganisms had occurred. Increased contact was associated with greater sharing between patients and therapy dogs, as well as between patients. A topical chlorhexidine-based dog decolonization was associated with decreased microbial sharing between therapy dogs and patients but did not significantly affect sharing between patients. These data suggest that the therapy dog is both a potential source of and a vehicle for the transfer of microorganisms to patients but not necessarily the only source. The relative contribution of other potential sources (e.g., other patients, the hospital environment) should be further explored to determine their relative importance.

## 1. Introduction

Animal-assisted intervention (AAI) therapy, the involvement of animals in alternative or complementary treatment, can improve the physical, mental, and social functions of patients within the healthcare setting. AAI has been widely implemented in a range of physio-social conditions in various settings in healthcare facilities and is becoming increasingly popular, especially for pediatric patients. AAI is an umbrella term that encompasses animal-assisted therapy (programs with a set clinical outcome), animal-assisted education (programs with a set educational/developmental outcome), and animal-assisted activities (programs without a pre-specified outcome) [1,2]. The most commonly reported patient benefits include a reduction in patients’ requirement for pain medication, enhanced socialization, and reduced stress and anxiety [3,4,5,6]. Conversely, the potential risks of incorporating animals into a hospital setting, where patients with decreased immune function are treated, must be considered. Nosocomial transmission of infectious disease agents, such as methicillin-resistant *Staphylococcus aureus* (MRSA), is a serious problem exacerbated by close contact and antimicrobial selective pressure inherent to healthcare settings, and we were concerned that dogs involved in AAI—for this paper, termed therapy dogs—may serve as mechanical vectors of transmission. Many common nosocomial pathogens, including MRSA and *Clostridioides difficile*, are zoonotic, in that they can be carried on animals and transmitted to humans [7,8,9]. While it is clear that therapy dogs can carry common hospital-associated pathogens [10,11,12], evidence is lacking on whether dogs transmit these microbiota to patients.

Microbes, including pathogens, function in the context of a more global microbial community, and other non-pathogenic microbiota may similarly be transmitted during these AAI sessions. Specifically, dogs have unique compositions of their nasal, dermal, and gastrointestinal microbial communities [13,14,15], which, compared to humans, could result in a distinct ability to acquire, carry, and/or spread hospital-associated pathogens. These distinct microbial communities could also uniquely influence the microbial composition of individuals that interact with the dogs, in a way that is fundamentally different from contact with other people or objects in the environment. This circumstance is best illustrated by data demonstrating the microbial shifts in humans resulting from pet ownership; pet owners often have more diverse microbial compositions that are more frequently shared between them [16,17]. Early-life pet ownership is associated with decreased incidence of immune dysfunction, and exposure to diverse microbes from farm environments, including animals, is protective against the development of asthma in children [18,19,20,21]. However, these studies focus on chronic exposure from living with animals and pets. It is uncertain if these same microbial shifts will occur with transient exposure of patients to a therapy dog, which, in our setting, was often less than one hour.

This study aimed to explore the potential for microbial sharing between pediatric patients, therapy dogs, and the hospital environment during animal-assisted intervention programs. We hypothesized that therapy dogs could serve as intermediary mechanical vectors in the transmission of microbes between the hospital environment and patients, and interaction with the therapy dog would increase patients’ risk of microbial exposure (Figure 1). We further examined whether the level of contact between patients and therapy dogs modifies this microbial sharing. This study used a topical antiseptic treatment on the therapy dog as a targeted intervention to mitigate potential risks from exposure to infectious agents to patients participating in AAI. We secondarily hypothesized that this topical disinfectant, aimed at decreasing the bacterial colonization in the therapy dog, would have downstream effects on microbial composition in patients. Improving our understanding of microbial dynamics that occur during an AAI session will contribute to our knowledge base regarding human–animal microbial exchange research in a novel setting and have practical implications for AAI program implementation.

## 2. Methods

### 2.1. Experimental Design and Sample Collection

This study was conducted at a pediatric oncology outpatient unit in a mid-Atlantic hospital between July 2016 and May 2017. The study protocol was approved by all applicable institutional review boards, institutional animal care and use committees, and scientific review committees prior to data collection. The therapy dog and handler were scheduled for one hour in a shared space, during which multiple patients interacted with the dog at the same time. Microbial samples were collected from the nasal mucosa of pediatric patients and therapy dogs with a sterile flocked swab (Puritan, Guilford, ME, USA) before and after the AAI visit, as well as the shared floor space with a vacuum dust filter [22]. Trained research staff performed all sample collection. During the visit, we observed interactions of the study participants with the dog, recording the total duration and frequencies of certain behaviors (petting, hugging, etc.). Blank sterile flocked swabs were collected at every visit as a negative control. Sample swabs were stored at −80 °C until processing.

The therapy dog team, consisting of the dog and its handler, completed two observational control visits abiding by established hospital protocol, then crossed over to two intervention visits with modifications to the hospital therapy dog protocol, as shown in Appendix A. Prior to the first intervention visit, the handler was given a 4% chlorhexidine-based veterinary prescription shampoo (DUOXO Ceva, Libourne, France) to use 24 h before the study visit. During the therapy visit, the dog was wiped down along the head and back, the “petting zone”, with 3% chlorhexidine wet cloths (DUOXO Ceva, Libourne, France) every 5 to 10 min.

### 2.2. Laboratory Processing

#### 2.2.1. 16SS rRNA Gene Amplification and Sequencing

Sample swabs and vacuum filter dust were thawed prior to DNA extraction. DNA sequencing was performed as previously described [16]; see also Appendix A for additional details. For each set of extractions, one blank swab exposed to laboratory air was processed as a negative laboratory control. Prior to sequencing, the total DNA concentration was obtained from a Qubit instrument, and the 16S rRNA gene copies per unit DNA were evaluated using quantitative PCR. The V1-3 region of the 16S rRNA gene was amplified using barcoded primers (27F, 534R) for the Illumina platform as previously described [23]. Sequencing was performed on the MiSeq instrument (Illumina, San Diego, CA, USA) using 300 base paired-end chemistry at the University of Pennsylvania Next Generation Sequencing Core. Microbial Mock Communities B (Even-Low v5.1, BEI Resources, NIAID NIH HMP) were amplified and sequenced as positive controls.

#### 2.2.2. Bioinformatics and Quality Control

QIIMEv2.7 was used for paired-end read assembly and quality filtering for the sequences from all samples [24]. The DADA2 plug-in for QIIME2.7 was used to remove chimeric sequences and sequences greater than 300 bp in length and to cluster sequences into amplicon sequence variants (ASVs) [25]. ASVs were matched to phylogeny using the mafft program for multiple masked sequence alignment [26] and FastTree to generate a phylogenetic tree from the masked alignment [27]. Taxonomy assignment used a naïve Bayes classifier [28] that was trained on our dataset (trimmed to 300 bp and matched to our primers), applying Greengenes v13.8 99% OTU match [29]. Taxonomic classification was confirmed by comparing the identification of the known Mock Community samples. For quality control purposes, suspected contaminants were identified and removed from the resulting feature table using the ‘decontam’ R package, based on the prevalence of taxa in the negative controls and the frequency of taxa as a function of the total DNA concentration and the 16S rRNA copies from qPCR [30]. Contaminants were identified independently at each processing step (field sampling, DNA extraction, and sequencing) and were sequentially removed. Information on the sequencing library and quality control measures can be found in Appendix A.

### 2.3. Statistical Analysis

Statistical analysis was performed in RStudio v1.1.423 [31]. To maintain the maximum number of samples for comparison, the sequencing data were not rarefied for statistical analysis [32,33]. Taxa tables, and matching phylogeny and taxonomy, were analyzed using the phyloseq pipeline to calculate alpha and beta diversity metrics [32]. The primary analysis was the change in microbial composition comparing pre- and post-visit overall and by host (human and dog), then stratifying by contact level and visit type (control versus intervention). Differential abundance of specific taxa between groups was analyzed using DESeq2 [34]. The Kruskal–Wallis nonparametric one-way analysis of variance test examined differential alpha diversity between all groups, and the Wilcoxon rank-sum test was used for pair-wise comparisons between groups; both tests were adjusted for multiple comparisons using the Benjamini–Hochberg false discovery rate (FDR) correction. To test which factors were most important in determining microbial composition, analyses were performed using the non-parametric permutational multivariate analysis of variance (PERMANOVA) with weighted and unweighted UniFrac distance metrics.

## 3. Results

### 3.1. Study Population and Samples

A total of four dogs were studied over 13 AAI visits (2–4 visits per dog team), with five (38%) being intervention study visits. Forty-five unique pediatric oncology subjects enrolled in the study (Table 1), with a mean age of 11.7 years old (SD 4.7). Four participants re-enrolled in the study, resulting in data from 49 study participants. Each therapy visit had a mean of 3.8 participants (SD 1.4, range 2–6). Thirty-nine participants (79.6%) reported having a pet at home, with 30 (61.2%) having a dog.

Individual contact behaviors and total patient–dog interaction times are presented in Appendix A. The frequency of key behaviors and total time spent with the therapy dog were aggregated to create an ordinal contact score. The median contact score was used as a threshold to create a binary contact level of “High” or “Low” contact. Fifty-one percent of patients were classified as “High” contact, and this was evenly distributed across control and intervention visits.

A total of 129 sample swabs were collected for microbial analysis (Table 1)**.** An additional 33 samples were processed for microbiome quality control. Swab samples were not collected from eight participants due to either the patient’s fear of the swabbing process or scheduling conflicts. Two subjects did not have pre-visit culture swabs collected, and three subjects did not have post-visit swabs collected.

### 3.2. Relative Abundance

The abundance of microorganisms differed across both host and sample location; Figure 2A shows the percent relative abundance of the top 25 most abundant genera. Certain bacterial species had significantly different abundance when comparing across sites, including *Staphylococcus* species in the nasal samples of both pediatric subjects and dogs (Figure 2). Subject and dog nasal samples had similar microbial compositions, with *Staphylococcus* species being dominant, but dog nasal samples had a greater abundance of *Moraxella* compared to the subjects’ greater abundance of *Streptococcus*. These data are summarized in Appendix A.

The degree of alteration of patients’ microbial communities varied with contact level and visit type (Figure 2b,c). Within control visits, subjects with low contact had a higher abundance of *Streptococcus* species after the visits compared to before. Subjects in control visits with high contact had a higher abundance of *Moraxella* species, both pre- and post-visit, compared to low-contact subjects in control visits. In contrast, there was no difference in the abundance of any genera between pre- or post-visit samples in high-contact subjects. Within the intervention visits, both high- and low-contact subjects had greater abundance of *Streptococcus* species before the visit and greater abundance of *Staphylococcus* species after the visit, specifically *S. epidermidis* and not *S. aureus* (Appendix A).

### 3.3. Alpha Diversity

Alpha rarefaction curves are presented in Appendix A. Alpha diversity significantly differed between hosts (humans versus dogs versus environment), as measured by the observed total taxa, Shannon, and Faith’s Phylogenetic metrics (Wilcoxon rank-sum test *p* < 0.001). This was a consistent observation when stratifying by pre- or post-visit and by control or intervention visit type.

When examining individual-level changes in alpha diversity that occur during a therapy visit, in high-contact subjects, there was an overall increase in within-sample diversity during control visits and an overall decrease during intervention visits; either no difference or the opposite difference occurred in low-contact subjects (Figure 3A,B). The change in alpha diversity between pre- and post-visit samples was significantly different in control versus intervention visits in high-contact patients when measured with Faith’s metric (Kruskal–Wallis *p* < 0.05), but not the Shannon metric or observed total taxa. A similar significant effect could be seen in therapy dog samples when using Faith’s metric (Kruskal–Wallis *p* < 0.01), with an overall increase in alpha diversity following control visits and a decrease following intervention visits (Figure 3D–F).

### 3.4. Beta Diversity

#### 3.4.1. Beta Diversity Distribution

Appendix A shows the overall distribution of samples in principal coordinate analysis plots for both unweighted and weighted UniFrac beta diversity metrics, by hosts (pediatric subjects, dog, or hospital environment), site, and pre- and post-visit status. Loose clustering was observed by host and sample site, but not by sample timing (pre vs. post). Clustering was also not observed by individual subject or visit date. Overall, the axes accounted for a maximum of 7.8% variation in unweighted UniFrac and 33.5% variation in weighted UniFrac.

#### 3.4.2. Beta Diversity Distance

Pediatric subjects were more similar to other subjects after the visits, as evidenced by their reduced microbial composition beta diversity distance (PERMANOVA pre vs. post FDR-*p* < 0.001). Patients were also more similar to therapy dogs after the visits (PERMANOVA pre vs. post FDR-*p* < 0.001). See example calculations in Appendix A, and results in Appendix A.

Subjects with high contact were more similar to other subjects (Figure 4A) and to the therapy dog (Figure 4C) after the visits, than to low-contact subjects (unweighted UniFrac metric PERMANOVA FDR-*p* = 0.0001–0.0003). The same pattern was observed in both control and intervention visits. Using a weighted UniFrac metric, high-contact subjects were more similar to other subjects in control visits (*p* = 0.0005) but not in intervention visits (Figure 4B). The reverse trend was observed between patients and the dog, with both high- and low-contact patients more similar in microbial composition to the therapy dog following intervention visits (*p* = 0.0001, 0.0005) but not control visits (Figure 4D).

## 4. Discussion

This study explored microbial transmission among pediatric oncology subjects and therapy dogs in a hospital-based AAI program. This study is the first to report on sampling patients, therapy dogs, and the hospital environment before and after a group AAI session, and the first to explore microbial community dynamics in this setting. Our data suggest that microbial sharing occurred during the AAI sessions, as microbial compositions of subjects were altered, both in overall diversity levels and in relative abundance of specific taxa. We further explored the effect of contact level between patients and therapy dogs on the alteration of nasal microbial communities following visits, and logically found that higher contact was associated with increased sharing between subjects and therapy dogs, and among subjects. Finally, we determined that an antiseptic decolonization intervention targeted to the therapy dog modifies the association between contact level and microbial sharing between therapy dogs and subjects, and between subjects as well.

### 4.1. Distinct Microbial Profiles and Shifts in Patients and Therapy Dogs

Patients, therapy dogs, and the hospital environment had distinct microbial communities, as evidenced by differences in the relative abundance of key species, differences in alpha diversity, and unique clustering of microbial composition in beta diversity. Human and dog nasal sites tended to be dominated by a few taxa at relatively high abundance (namely *Staphylococcus*, *Streptococcus*, and *Moraxella*) and had distinct beta diversity clusters on PCoA plots. These data are confirmed by other studies that have evaluated the microbiome of human skin and nasal samples [35,36,37,38].

We observed microbial community shifts in pediatric subjects and therapy dogs during an AAI therapy session. This was demonstrated by the increase in within-sample alpha diversity levels in subjects and dogs, more similar microbial compositions between groups following the visits, and changes in the relative abundance of certain taxa, specifically *Staphylococcus*. Beta diversity distance, represented by both unweighted and weighted UniFrac metric, was calculated as the difference in beta diversity between subject samples and between subject and dog samples. We then used these metrics as a proxy for the degree of microbial sharing between these hosts, as these metrics best indicated shifts in the microbial community structure that occurred during the visits. From these data, we found that subjects had more similar nasal microbial structure to other subjects and therapy dogs after the therapy visit compared to before, suggesting that sharing of microbiota occurred. Such sharing has been demonstrated in other human–animal microbiome studies, particularly those that evaluated pets in the home [16,17].

### 4.2. Closer Contact between the Patient and Therapy Dog Increased Microbial Sharing

Our data also suggest that the patient–dog contact level modifies microbial sharing between subjects and therapy dogs, as well as between subjects. While contact level was primarily an indicator of the degree of interaction between a subject and a therapy dog, by extension, it can also reflect the degree of contact that occurs between a patient and the hospital environment, other patients, and other aspects of the therapy visits (model shown in Figure 1). In other words, a subject with a high contact score will have higher contact with the therapy dog, as well as with other patients and individuals, including the therapy dog handler, and with the hospital environment. Thus, it is logical that high contact with the therapy dog suggests higher contact with the more general environment, and this was reflected in our data as being positively associated with increased microbial sharing.

Our data suggest that high-contact patients with more interaction with various aspects of the therapy programs shared more microbes both with other patients and with the therapy dogs. This was demonstrated by an increase in within-sample alpha diversity and more similar beta composition between samples. Interestingly, there were differences using phylogenetically weighted versus unweighted metrics. Faith’s Phylogenetic alpha diversity changes were greater than Shannon alpha diversity, suggesting that more phylogenetically distinct microbiota are driving the increased alpha diversity in subject samples. Our unweighted UniFrac distance appeared to show stronger microbial sharing between high-contact patients and therapy dogs, and among high-contact patients, while the phylogenetically weighted UniFrac distance appeared to show significant sharing of rare taxa among high-contact subjects, but not between humans and dogs.

Taken together, these data suggest that bacteria are shared among humans, and between humans and dogs, in the AAI setting, but rare bacteria are less commonly shared between humans and dogs. Our PCoA distributions and relative abundance results, in addition to previous studies on pet dogs [14,17,39,40], have shown that dogs have distinct microbial communities compared to humans. These differences could possibly drive the differences that we observed in weighted beta metrics comparing subject-to-dog composition difference to subject-to-subject composition difference. These data therefore support the hypothesis that dogs can serve as intermediary vectors in the spread of human-origin common microbiota between patients, but may not be sharing their own unique microbiota with patients. However, significant sharing of rare taxa occurred among subjects in the AAI setting. These additional data suggest that the therapy dog is only one potential pathway by which microbes can be transmitted during these group AAI therapy sessions, with other pathways shown in Figure 1 potentially being more influential.

### 4.3. Canine Decolonization Intervention Modified Microbial Sharing

We tested a novel application of a topical chlorhexidine to therapy dogs and assessed how this canine decolonization intervention influenced microbial sharing between patients and dogs and among patients. Our data preliminarily demonstrate that the dog microbial decolonization intervention modifies the observed relationship between contact level and microbial sharing. The decolonization intervention appeared to have influenced more phylogenetically distinct, rare taxa, as different outcomes were obtained using phylogenetically weighted versus phylogenetically unweighted diversity models. Within the intervention visits, microbial sharing of common taxa was still observed among subjects and between subjects and therapy dogs, as evaluated by the unweighted UniFrac distances. However, unlike in control visits, the weighted UniFrac distances suggest that rare taxa were not shared among subjects. Thus, the intervention appears to have blocked the sharing of rare, phylogenetically diverse taxa between humans. High-contact subjects had more significantly decreased alpha diversity levels following intervention visits than in control visits, indicating that our canine-centered decolonization had indirect effects on the microbial diversity levels of human subject samples. These data are consistent with previous data on the effects of the hospital built-environment microbiome on patient microbial composition, and particularly those data regarding the influence of environmental cleaning regimens on patient microbiota [41,42,43].

Interestingly, following intervention visits, both high- and low-contact patients appeared to have more similar microbial compositions to therapy dogs, using the phylogenetically weighted UniFrac metric. This contrasts control visits, where subjects of both contact levels had less similar microbial communities to dogs. This difference is explained less by microbial sharing, as the effect of the disinfectant intervention on the dog’s microbiota. The decolonization selectively removes unique dog taxa from the dog itself, perhaps more easily allowing recolonization with the microbial community of the subjects.

The intervention was also associated with changes in the abundance of specific taxa. High-contact patients had a higher relative abundance of staphylococcal species following intervention visits compared to high-contact patients following control visits. This change was primarily driven by *S. epidermidis*, a predominant human nasal and skin commensal, rather than *S. aureus*, which can be more pathogenic. Because this metric compares relative rather than absolute abundance within each sample group, it is not surprising that human commensals are of greater relative abundance in intervention visits than control visits, since the subjects were exposed to fewer taxa from the therapy dog.

Overall, while the intervention influenced microbial composition, diversity levels, and sharing among humans, it primarily exerted these effects by modulating the therapy dog’s microbial composition. If the therapy dog was the only or primary source of microbes that were transferred to patients during AAI sessions, we would expect to see reduced sharing of both common and rare taxa between patients and dogs following intervention visits when the therapy dog pathway is blocked. Since we only see this pattern with phylogenetically distinct, rare taxa, not common taxa, it appears more likely that the therapy dog serves as an intermediary point of microbial sharing, rather than a source of microbes. Thus, the dog is only one of many possible pathways of microbial sharing (Figure 1), and these other pathways may contribute more to microbial changes seen in subjects attending AAI visits.

### 4.4. Strengths, Limitations, and Future Directions

While designed as a pilot study to assess feasibility, this study expressly targets microbial transmission that occurs during hospital AAI programs and is the first to report on sampling multiple components before and after each visit. As such, these data provide a critical foundation for larger studies in this area. Previous studies have focused exclusively on carriage in the therapy animal or have assessed aggregated rates of infection diagnosis in departments with or without AAI programs [11]. By sampling multiple components—the patients, the therapy dogs, and the hospital environment—we can begin to elucidate exposure pathways from these individual data points. In addition to our novel sampling strategy, this is the first study to assess the effect of patient–dog contact on microbial sharing. Human–animal contact level has been previously described as a risk factor for the exposure and acquisition of pathogens in the case of pet ownership [44,45]. However, it was unknown if the same positive association would occur with the transient contact between patients and therapy dogs. This study also benefits from the novel deployment of an established canine decolonization procedure, adapted from veterinary clinical protocols for canine patients with dermatopathologies. The intervention appeared to limit the spread of the therapy dog’s own unique microbiota to patients, and also reduced the therapy dog’s role as an intermediary vector in the spread of microorganisms between patients or other individuals and the hospital environment.

This study does have practical limitations. As a feasibility study in preparation for a larger infection control trial, including the implementation of the decolonization intervention, it is limited by the small sample size, particularly when considering the number of unique dogs. While our sampling was fairly extensive, sampling other sites, both on the subjects and in the hospital environment, as well as other individuals, such as healthcare workers and the handlers, may have provided additional data to support alternative hypotheses. Multiple pathways depicted in Figure 1 are, in fact, quite challenging to examine, and blanket statements inferring directionality of transmission from therapy dogs to patients or vice versa should be taken with appropriate caution. Finally, this experiment assessed microbial exposure and composition at one time point. Our data do not address whether the changes observed during the visit will persist and, if so, for how long. These data also do not support claims regarding the health outcomes related to these microbial community shifts, particularly related to the exposure of potentially pathogenic microorganisms and to rare taxa from the therapy dog.

Future work on this topic will expand to studying AAI sessions that involve only one child per dog, thus providing more controlled insight into potential microbial transmission pathways and increasing the generalizability of findings to other situations. Studies that sample within different hospital departments with varying compositions of patients, and various hospitals, will also be required to increase the generalizability. Lastly, longitudinal studies are required to explore the temporal stability of these microbial shifts observed in patients and determine whether it leads to clinically significant outcomes. Such longitudinal studies are especially important when considering the exposure to rare dog taxa, given that early-life exposure to pets is associated with decreased incidence of allergic and atopic diseases in children [46,47], and having a diverse microbiome is protective against numerous health outcomes and can be protective against colonization from pathogens [48,49]. If such data suggest that exposure to therapy dogs, even briefly during AAI programs, can benefit microbial diversity and microbial community resilience over the longer term, this will be a previously undescribed benefit of AAI and may increase its utilization in patient care.

## 5. Conclusions

These findings indicate that, while there is presumed microbial sharing between pediatric patients and therapy dogs, and while the therapy dog has the potential to serve as an intermediary vector of microbial spread, other potential transmission pathways (patient-to-patient, and environment-to-patient) also appear to contribute to microbial sharing during group AAI visits. Our results also suggest that the therapy dog could be a source of more unique microbes to patients. As hospital exposure and certain therapies decrease microbial diversity in patients, therapy dog exposure may provide a novel way to mitigate this imbalance and transmit potentially beneficial microorganisms that could be protective against hospital pathogen colonization and infection. This study shows that microbial community alterations in patients and therapy dogs during these therapy programs warrant additional research, which will make these programs safer and more sustainable.

## Figures and Tables

**Figure 1 microorganisms-09-01054-f001:**
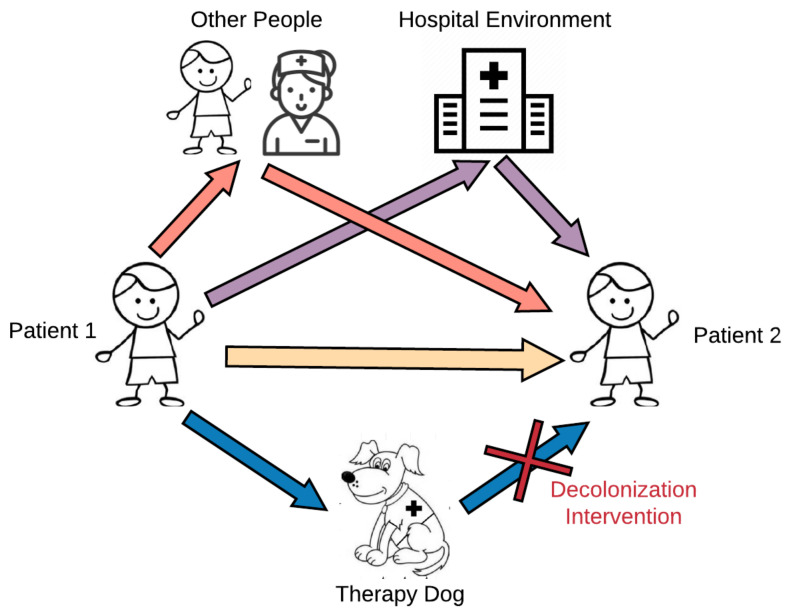
Microbial pathways during animal-assisted intervention programs.

**Figure 2 microorganisms-09-01054-f002:**
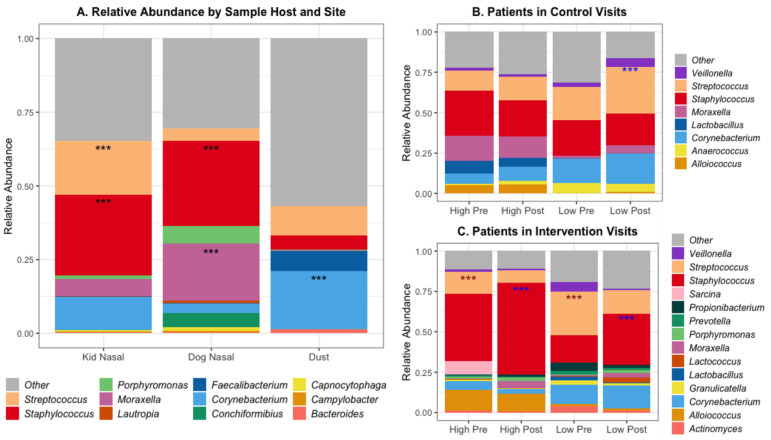
Relative abundance of top 20 genera by (**A**) Sample Host and Site, and within (**B**) Patients in Control Visits and (**C**) Patients in Intervention Visits. *** Benjamini–Hochberg adjusted *p*-values < 0.001 for differential abundant genera using a negative binomial model (DESeq) between sample sites. Within patients (**B** and **C**): **blue ***** = higher in post samples, **red *** **= higher in pre samples, Mean total DNA concentration in patients in control = 6.28, in intervention = 4.42 (ng/uL), Mean qPCR 16S gene copies in patients in control = 22,254, in intervention = 8691 (/uL DNA).

**Figure 3 microorganisms-09-01054-f003:**
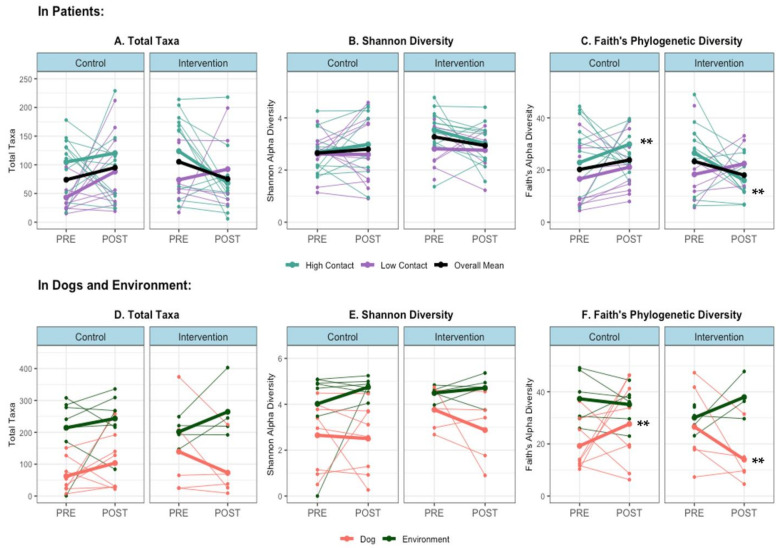
Alpha diversity by sample host and site, and within patient samples (**A**) Total Taxa in Patients (**B**) Shannon Diversity in Patients (**C**) Faith’s Diversity in Patients (**D**) Total Taxa in Dogs and Environment, (**E**) Shannon Diversity in Dogs and Environment and (**F**) Faith’s Diversity in Dogs and Environment. Thin lines = within-subject changes, bold lines = aggregated group means, ** Kruskal–Wallis test *p* < 0.05 for median difference in change in alpha diversity level (post-pre) in control vs. intervention (in high-contact patients and dogs).

**Figure 4 microorganisms-09-01054-f004:**
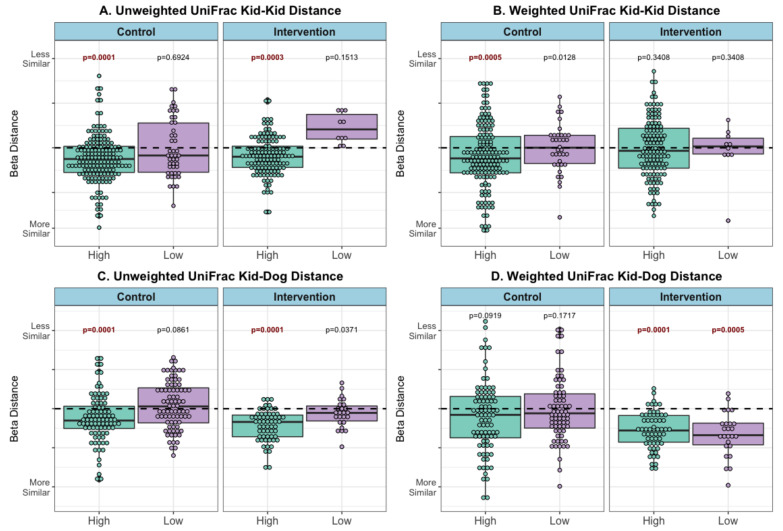
Beta distance for microbial composition difference, by contact level and visit type (post–pre visit) (**A**) Unweighted UniFrac Distance Kid to Kid (**B**) Weighted UniFrac Distance Kid to Kid (**C**) Unweighted UniFrac Distance Kid to Dog, and (**D**) Weighted UniFrac Distance Kid to Dog. PERMANOVA model *p*-value results for difference in microbial composition beta distance between patients pre- compared with microbial composition beta distance between patients post-visit (kid-kid) or difference in microbial composition beta distance between patients and therapy dogs pre- compared with microbial composition beta distance between patients and therapy dogs post-visit (kid-dog), within each stratification (visit type and contact level). Refer to Appendix A for example calculations and pre/post distances, **bold ** FDR-corrected *p* < 0.005.

**Table 1 microorganisms-09-01054-t001:** Study population and samples.

	All Visits	Control Visits	Intervention Visits
**Study Population**			
**Patients**		N (% Total)	N (% Total)
N total sampled	49 *_45_	26 (53%) *_23_	23 (47%) *_22_
Male (%)	31 (63%)	15 (58%)	16 (69%)
Age (y), mean (range)	11.68 (1.9–20.4)	11.07 (1.9–18.4)	12.41 (3.5–20.4)
High Contact (%)	25 (51%)	12 (46%)	13 (56%)
**Visits**		N (% Total)	N (% Total)
Total	13	8 (62%)	5 (38%)
Patients per visit, mean (range)	3.77 (2–6)	3.25 (2–5)	4.6 (3–6)
**Therapy Dogs**			
N Unique Dogs	4		
Male (%)	1 (25%)		
Age (y), mean (range)	6.43 (1.5–12)		
**Samples**			
From Patients	79	43 (54%)	36 (46%)
From Dogs	26	16 (62%)	10 (38%)
From Environment	24	14 (58%)	10 (42%)
**Total Samples**	129	73 (57%)	56 (43%)
Field Blanks	12	7 (58%)	5 (42%)
Laboratory Controls	21		
**Total Controls**	33		

* 45 patients with microbial samples collected, 23 in control and 22 in intervention.

## Data Availability

The dataset supporting the conclusions of this article is available in the NCBI Sequence Read Archive (SRA) database under BioProject PRJNA695069, BioSample SAM17600695. Unix and R code used for analysis can be found under KRD’s Github repository: https://github.com/kathryndalton/AAT_pilot_analysis (accessed on 7 April 2021).

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
