# Peer review of "Microbial Sharing between Pediatric Patients and Therapy Dogs during Hospital Animal-Assisted Intervention Programs"

_microorganisms, 2021, doi:10.3390/microorganisms9051054_

Round 1
Reviewer 1 Report
Dear Authors, you have made excellent work.
The study is accurate and detailed in the experimental design, it is full of additional information useful to reach even less experienced readers.
However, I believe that some definitions need to be changed since for example an animal should never be "used" but "involved" in an Animal-Assisted Intervention.
There is a large international literature that defines and clarifies the areas of application and the purposes of these interventions (Animal Assisted Therapy - AAT, Animal Assisted Activity - AAA, Animal Assisted Education - AAE), therefore the definitions you report should be revised.
If the animal species is the dog, it would be appropriate not to define it as a "therapy animal" but rather a "co-therapist dog".
These are only suggestions that I would like to share with you, as I anticipate that your study will be highly rated and widely disseminated, so it would be wise to make the changes indicated.
Finally, I verified that the manuscript was not written following the instructions for the authors, as well as all the references both in the text and in the appropriate section. Please check carefully.
Reviewer 2 Report
Reviewer’s Comments
Name of journal: Microorganisms (ISSN 2076-2607)
Manuscript ID: microorganisms-1195111
Type: Article
Title: Microbial Sharing between Pediatric Patients and Therapy Animals during Hospital Animal-Assisted Intervention Programs
Authors: Kathryn R. Dalton * , Kathy Ruble , Laurel E. Redding , Daniel O. Morris , Noel T. Mueller , Roland J. Thorpe, Jr. , Jacqueline Agnew , Karen C. Carroll , Paul J. Planet , Ronald C. Rubenstein , Allen R. Chen , Elizabeth A. Grice , Meghan F. Davis
Reviewer
Summary:
This study investigates the microbial sharing between humans and animals that occurs within the health setting during animal-assisted intervention programs to address important concerns regarding the potential transmission of pathogens that limits the use of the program. . This study evaluated the potential for microbial sharing between pediatric patients and therapy dogs through the characterization of microbial communities with 16S rRNA sequencing. Both patients and animals exhibited changes in the relative abundance and overall diversity of their nasal microbiome, suggesting that an exchange of microorganisms had occurred. Greater contact was associated with greater sharing between patients and 40 therapy animals and between patients. These data suggest that the therapy animal is both a potential source and a vehicle for the transfer of microorganisms to patients, but not necessarily the only source. The text is very well written and focuses on a very delicate and important clinical topic. The topic is suitable for the magazine in which it was submitted. I recommend its publication.
Minor issues:
1) The introduction should be enriched with a part dealing with potentially pathogenic species causing zoonoses.
2) Only the upper respiratory tract microbiota was evaluated. It could also be very interesting to evaluate the skin microbiota, before and after contact with therapy animals, also as a function of the decolonization treatment.
3) In Figure 2, increase the font size of the genus and in italics
4) In Line 221: the graph shows a variation in the abundance of the genus Moraxella before and after the visit, not reported in the text. Describe this in the text, and include it in the results.
